# Tumors Resistant to Checkpoint Inhibitors Can Become Sensitive after Treatment with Vascular Disrupting Agents

**DOI:** 10.3390/ijms21134778

**Published:** 2020-07-06

**Authors:** Michael R. Horsman, Thomas R. Wittenborn, Patricia S. Nielsen, Pernille B. Elming

**Affiliations:** 1Experimental Clinical Oncology-Department of Oncology, Aarhus University Hospital, DK-8200 Aarhus, Denmark; Wittenborn@biomed.au.dk (T.R.W.); pernille.elming@oncology.au.dk (P.B.E.); 2Department of Pathology, Aarhus University Hospital, DK-8200 Aarhus, Denmark; pariniel@rm.dk

**Keywords:** vascular disrupting agents, combretastatin, OXi43503, checkpoint inhibitors, anti-PD-1, anti-PD-L1, anti-CTLA-4, C3H mammary carcinoma, tumor growth delay

## Abstract

Immune therapy improves cancer outcomes, yet many patients do not respond. This pre-clinical study investigated whether vascular disrupting agents (VDAs) could convert an immune unresponsive tumor into a responder. CDF1 mice, with 200 mm^3^ C3H mammary carcinomas in the right rear foot, were intraperitoneally injected with combretastatin A-4 phosphate (CA4P), its A-1 analogue OXi4503, and/or checkpoint inhibitors (anti-PD-1, PD-L1, or CTLA-4 antibodies), administered twice weekly for two weeks. Using the endpoint of tumor growth time (TGT5; time to reach five times the starting volume), we found that none of the checkpoint inhibitors (10 mg/kg) had any effect on TGT5 compared to untreated controls. However, CA4P (100 mg/kg) or OXi4503 (5–50 mg/kg) did significantly increase TGT5. This further significantly increased by combining the VDAs with checkpoint inhibitors, but was dependent on the VDA, drug dose, and inhibitor. For CA4P, a significant increase was found when CA4P (100 mg/kg) was combined with anti-PD-L1, but not with the other two checkpoint inhibitors. With OXi4503 (50 mg/kg), a significant enhancement occurred when combined with anti-PD-L1 or anti-CTLA-4, but not anti-PD-1. We observed no significant improvement with lower OXi4503 doses (5–25 mg/kg) and anti-CTLA-4, although 30% of tumors were controlled at the 25 mg/kg dose. Histological assessment of CD4/CD8 expression actually showed decreased levels up to 10 days after treatment with OXi4503 (50 mg/kg). Thus, the non-immunogenic C3H mammary carcinoma was unresponsive to checkpoint inhibitors, but became responsive in mice treated with VDAs, although the mechanism remains unclear.

## 1. Introduction

When tumors arise, it is because they have successfully evaded the immune system [1]. For tumor cells to remain viable and for the tumor to continue to grow, they, like normal cells, require an adequate supply of oxygen and nutrients. Initially these factors come from the host normal vascular supply, but when tumors reach a size of a few millimeters the cells can exceed the diffusion distance of oxygen and other essential nutrients [2]. Further tumor development is then only possible if the tumor forms its own functional vascular system from the host vessels by the process of angiogenesis [3]. Unfortunately, this neo-vasculature is unable to meet all the demands of the growing tumor mass and regions of low oxygenation (hypoxia) and nutrient deficiency, develop [4]. Although tumor cells can tolerate these extreme environmental conditions for some time, as the tumor continues to grow these deprived cells eventually die. According to the “cancer immunity cycle” [5], cell death in tumors releases cancer antigens that are identified by antigen presenting cells (APCs). These APCs prime and activate T-cells in the lymph node, and these T-cells then traffic to the tumor, infiltrate and recognize the tumor, and thereby kill additional tumor cells [5]. However, ligands for immune checkpoint proteins (ICPs) have developed to prevent over-activation of the normal immune system; activated T cells display ICPs as a physiologic response against prolonged immune system activation that may potentially damage normal tissues [6]. These ICPs also allow the tumor to avoid the immune system [7]. The two most studied ICPs are the cytotoxic T-lymphocyte-associated protein 4 (CTLA-4), and programmed death 1 (PD-1) [8]. The ligand of CTLA-4, B7, is a protein expressed on activated antigen presenting cells, while for PD-1 it is the programmed death 1 ligand (PD-L1), found on a variety of immune cells, as well as expressed in several tumor types [5].

The development of various antibodies against these checkpoints, allows the immune system to bypass the blocks, and a number of so-called checkpoint inhibitors have undergone testing in a variety of cancer sites, especially melanoma and lung [9,10]. These studies and others have resulted in unprecedented improvements in outcome in patients with a spectrum of solid tumors, and has generally established immunotherapy as the fourth modality in cancer treatment after surgery, radiation, and chemotherapy. However, despite significant improvements in patient outcome many patients actually do not respond to immunotherapy [11,12]. Consequently, considerable efforts are underway to find alternative treatments that, combined with checkpoint inhibitors, could improve patient response.

One clinically relevant approach to this issue is to combine checkpoint inhibitors with radiation. While radiation directly kills tumor cells, it also has the potential to induce the production of damage associated molecular patterns (DAMPs), and thereby indirectly induce an immune response [13]. However, although there is clear evidence that the immune system plays a critical role in influencing the degree of local control of irradiated tumors [14], radiation itself does not appear to induce any anti-tumor responses outside the radiation field. This so called abscopal effect is what one would expect if radiation actually induced an immune response, and a recent systematic analysis of the number of the reported cases on the abscopal effect in patients receiving radiation therapy suggested this to be an extremely rare clinical event [15]. The immune checkpoints could be responsible for this negative finding, so combining radiation with checkpoint inhibitors could be beneficial. Indeed a number of pre-clinical studies [16,17,18] and even clinical studies [19,20,21] suggest this to be an effective combination for improving tumor response. However, there is no consensus as to the radiation dose/schedule or timing for the best effect when combined with checkpoint inhibitors. Furthermore, regions of hypoxia, which are a characteristic feature of both animal [22] and human [23] solid tumors, contain viable cells that are a known resistant factor for radiation [24,25,26]. Recent studies suggest that hypoxia can also decrease the efficacy of immunotherapy [27,28].

An alternative approach for effectively increasing tumor damage, and one that can effectively reduce the degree of tumor hypoxia, is the use of agents that target the tumor vascular supply. As described earlier, a functional tumor blood supply is critical for the survival of tumor cells and the growth and development of the tumor mass. This significance has led to the development of various vascular targeting agents, which either inhibit the angiogenesis process (angiogenesis inhibitors) or damage the already established tumor neo-vasculature (vascular disrupting agents; VDAs) [29,30]. These vascular targeting agents have undergone extensive pre-clinical evaluation [25,30] and many of them are currently in clinical development [31,32]. The aim of our current pre-clinical study was to investigate whether the leading small molecule VDA, combretastatin A-4 phosphate (CA4P; fosbretabulin; zybrestat), and the A-1 analogue OXi4503 [32] could influence the immune response of an established tumor model that is actually non-immunogenic.

## 2. Results

The effect of the various antibodies alone on the growth of this C3H mammary carcinoma are shown in Figure 1 and Figure 2. Regardless of whether using the anti-PD-1, anti-PD-L1, or anti-CTLA-4 antibody, there was absolutely no effect on tumor growth compared to that found in untreated controls (Figure 1A,B and Figure 2A). CA4P (100 mg/kg), administered twice weekly for two weeks, significantly increased the tumor growth time (TGT5; time for tumors to reach five times the starting treatment volume). Combining CA4P with the checkpoint inhibitors resulted in an increase in TGT5 for each VDA and inhibitor combination, but this additional effect was only statistically significant when CA4P and anti-PD-L1 were combined, not when the VDA was combined with the other two antibodies (Figure 1).

Multiple treatments with OXi4503 (50 mg/kg) also significantly increased TGT5 compared to controls (Figure 2A). When combined with anti-PD-1 the TGT5 values were the same as seen with OXi4503 alone, although one mouse in the OXi4503 + anti-PD-1 group did show tumor control at 90 days, but overall there was no significant difference compared to OXi4503 alone. For the combination of OXi4503 and anti-PD-L1, 60% of mice had TGT5 values that were in the same range as OXi4503 alone. The remaining 40% showed a small increase in TGT5. Combining OXi4503 and anti-CTLA-4 antibody showed that some 50% of mice had TGT5 values that were in the same range as the OXi4503 alone group, but that the remaining 50% had a greater TGT5; for two of these mice, the TGT5 values were extremely large. From a statistical standpoint, the TGT5 values for the combination of OXi4503 with either anti-PD-L1 or anti-CTLA-4 were significantly greater than for OXi4503 alone, with the significance levels being higher for the combination with anti-CTLA-4 (*p* = 0.003) than with anti-PD-L1 (*p* = 0.03). Figure 2B shows the effect of using lower OXi4503 doses in combination with anti-CTLA-4. All OXi4503 doses (5, 10, and 25 mg/kg) significantly increased TGT5 above that seen with untreated controls. When comparing the different OXi4503 doses, a dose-dependent increase in TGT5 was observed, that appears to be maximal at around a dose of 25 mg/kg, and the 25 and 50 mg/kg doses resulted in similar TGT5 values. Figure 2B also shows the effect of combining the three lower OXi4503 doses with anti-CTLA-4. We only used the anti-CTLA-4 antibody, rather than anti-PD-1 or anti-PD-L1, because it was the inhibitor that had the greatest effect with the higher 50 mg/kg dose (Figure 2A). However, when anti-CTLA-4 was combined with either the 5, 10, or 25 mg/kg OXi4503 doses, no further significant improvement was found, despite 30% of mice treated with OXi4503 (25 mg/kg) and anti-CTLA-4 resulting in complete tumor control.

In an attempt to shed some light on the potential mechanism involved, we monitored the expression of CD4+ and CD8+ in tumors. The results of the analysis of histological tumor sections at various days after injecting a single dose of OXi4503 (50 mg/kg) are summarized in Figure 3. Surprisingly, there was a rapid drop in both CD4+ and CD8+ expression levels within 1 day after injecting the VDA. Partial recovery was then observed at day 4 for CD4+ and complete recovery at day 3 for CD8+. However, for both parameters, another decline was then observed up to 10 days after treatment. For CD4+, all the values, except that at four days, were significantly different from controls. Despite finding similar trends with CD8+, none of the values were actually statistically different from controls.

Figure 4 shows the effect of the various treatments on mouse body weight to get an indication of whether any of these treatments induced toxicity. In control tumor bearing mice, body weight slowly decreased as a result of the increasing tumor size (see Figure 1 for tumor growth in the same animals), but even at the end of the observation period, this loss in body weight is less than 10%. Similar results were seen with each of the checkpoint inhibitors. With CA4P treatment there was a gradual decline in body weight over time, while with OXi4503 there was a rapid drop in body weight of around 5–10% one day after starting the treatment, but then the values fluctuate around this level for the next two weeks. Combining CA4P with anti-PD-L1, or OXi4503 with anti-CTLA-4 (these combinations were selected because they had the greatest anti-tumor effects as shown in Figure 1 and Figure 2) resulted in body weight values that were no different from those seen with the VDAs alone.

## 3. Discussion

The C3H mammary carcinoma model used in the current study arose spontaneously in a C3H mouse [33], but was later established in the more stable CDF1 mouse variant [34]. Implantation of tumor material into CDF1 mice typically resulted in a 100% take rate, with tumors reaching our standard treatment size of 200 mm^3^ generally within two-three weeks after challenge, and five times that volume within a further 7–10 days. This suggests a lack of any immune response for this tumor type in this mouse strain. Consequently, it was not a surprise to find that the growth of the C3H mammary carcinoma was not affected by any of the three checkpoint inhibitors at the dose (10 mg/kg) and schedule (twice/week for two weeks) used. Additional observations in which we gave the same 10 mg/kg dose for four consecutive days after tumors had reached 200 mm^3^ also had no effect on tumor growth (data not shown).

However, treating mice with either CA4P (Figure 1), or its analogue OXi4503 (Figure 2), caused this C3H mammary carcinoma to become responsive to specific checkpoint inhibitors. For both VDAs an enhanced effect was observed to some extent with all three antibodies. Although with CA4P a statistically significant increase was only seen when combined with anti-PD-L1, while for OXi4503 significant enhancements were found when it was given in combination with anti-PD-L1 or anti-CTLA-4, although the latter combination was probably greatest. Why we see these small differences between CA4P and OXi4503 is unknown, but differences in the mode of action of the two drugs may play a role. The mechanism of action of CA4P primarily involves a selective disruption of the cytoskeleton of proliferating endothelial cells, resulting in endothelial cell shape changes and subsequent thrombus formation and vascular collapse [35,36]. Our previous studies in the C3H mammary carcinoma confirm CA4P can significantly reduce tumor perfusion for extensive periods [37,38,39], and this leads to an induction of necrosis and corresponding tumor growth inhibition [37,40,41]. Although OXi4503 is an analogue of CA4P, it is superior to the parent compound [40] because it induces greater vascular damage [42] and undergoes oxidative activation to a quinine intermediate, thereby, also giving it direct cell killing properties [43]. Indeed, our own studies using this C3H mammary carcinoma clearly show OXi4503 to have a greater effect than CA4P on the induction of tumor necrosis and subsequent inhibition of tumor growth [40]. In fact, the effect of 100 mg/kg CA4P was less than that seen with either 25 or 50 mg/kg OXi4503 in terms of necrosis development [40] or tumor growth delay, as shown in the current study when comparing the TGT5 values in Figure 1 and Figure 2.

Whatever, the explanation for the different effects for CA4P or OXi4503 with the checkpoint inhibitors, there is still the issue of why we are able to convert an “immunological cold” tumor model into one that is “immunologically hot”. This may simply be that the significant cell killing induced by the VDAs causing a substantial elevation of antigens/DAMPs, which has the potential to increase the number of active T-cells [5]. However, the CTLA-4 checkpoint normally prevents this activation [5,6,7,8]. The use of an anti-CTLA-4 antibody could bypass this block. Yet, an enhanced anti-tumor effect was only seen when anti-CTLA-4 was combined with 50 mg/kg OXi4503, although to some extent with the 25 mg/kg dose; at this lower dose the difference between OXi4503 alone and OXi4503 + anti-CTLA-4 was actually just non-significant (*p* = 0.06−0.07), but 30% of mice in the combination treatment had complete tumor control. There was no effect combining anti-CTLA-4 with lower OXI4503 doses (5 and 10 mg/kg), nor the 100 mg/kg CA4P dose. Since we found that the anti-tumor effects of 25 and 50 mg/kg OXi4503 are somewhat similar, and superior to the lower OXI4503 doses (Figure 2) or the CA4P dose used (Figure 1), it might suggest that the enhancement seen in combination with anti-CTLA-4 only occurs when a specific level damage is obtained.

When using anti-PD-L1 an enhanced anti-tumor response was seen regardless of whether using CA4P (100 mg/kg) or OXi4503 (50 mg/kg) as shown in Figure 1 and Figure 2. We did not combine lower OXi4503 doses with anti-PD-L1. Since PD-L1 affects the interaction between T-cells and tumor cells [5], it suggests that the improved response to the combined VDA and anti-PD-L1 antibody treatment is mediated through some tumor cell related parameter. One potential candidate could be hypoxia. The presence of viable hypoxic cells is a characteristic feature of solid animal [22] and human [23] tumors. Such cells are a source of resistance to radiation and certain types of chemotherapy, as well as playing a major role in influencing malignant progression in terms of tumors being more aggressive and metastatic [24,25,26]. Hypoxia also seems to play a role in influencing anti-cancer immune responses [27,28]. In response to hypoxia, cells experience a number of molecular changes that are mediated by intracellular signaling pathways under the control of hypoxia inducible factors (HIFs) [44,45,46]. Via several of these HIF-dependent mechanisms, hypoxia can cause tumor resistance to immune attack [46,47,48,49]. It can promote an immunosuppressive microenvironment by recruiting protumor immune cells such as Tregs, tumor-associated macrophages, neutrophils, and myeloid-derived suppressor cells [44,45]. Hypoxia can also alter the function of immune cells and increase resistance of tumor cells to the cytolytic activity of immune effectors [47,48]. Immune checkpoints are also affected by hypoxia, causing an up-regulation of PD-L1 and increased expression of CTLA-4. [49,50]. Hypoxia also has indirect effects on immune response by increasing lactate levels, adenosine accumulation, and vascular endothelial growth factor expression, all of which can inhibit anti-tumor immunity [27,51].

The reduction in tumor perfusion induced by CA4P or OXi4503 results in substantial cell killing and necrosis induction [37,38,39,40,41,42] and those cells most likely to die first are those tumor cells already under reduced oxygen and nutrient conditions, namely the hypoxic cells. Studies with invasive oxygen electrodes [52,53,54], non-invasive positron emission tomography [54,55], and even radiation response [52,53] not only confirm that our C3H mammary carcinoma model contains significant regions of hypoxia, but that it accounts for some 10–20% of the total viable population [52,53]. Such hypoxic cells are resistant to radiation treatment and several studies have shown that combining VDAs with radiation is an effective combination, presumably due to the VDAs eliminating this resistant population [25,30,56,57,58]. However, for CA4P, the benefit was only with the drug administered after irradiating, not when CA4P was administered before the radiation [59]. This suggests that a population of normally radiation sensitive cells become hypoxic following the drug treatment and survive, thus becoming a source of resistance to any subsequent radiation treatment. However, this effect seems transient with the resistance disappearing within three days; the induced hypoxic cells either re-oxygenate or die if the oxygen and nutrient starvation are prolonged [59]. For OXi4503 this is not the case since the same enhancement of radiation response occurs regardless of whether administering the drug before or after irradiating [59]. This is probably because of direct cell killing of any induced hypoxic cells by the quinine intermediate. Clearly, additional studies are required to categorically demonstrate that the reduction in tumor hypoxia induced by these VDAs plays a role in influencing the immune response of our C3H mammary carcinoma. Support for this idea comes from another pre-clinical study in which tumor hypoxia was reduced by allowing animals to breathe high oxygen content gas [48]. That study demonstrated that tumor bearing mice exposed to an oxygen environment of 60% showed an inhibition of tumor progression, a decrease in metastatic spread, and prolonged survival, when compared to mice maintained under normal atmospheric conditions (21% oxygen). Mechanistic studies revealed that this hyperoxia resulted in a decrease in tumor hypoxia, an increase in proinflammatory cytokines, a decrease in immunosuppressive molecules, and a reduced immunosuppression by regulatory T-cells [48].

Any increase in T-cell activation or activity resulting from a reduction in hypoxia in our C3H mammary carcinoma might have been reflected by an increase in T-cell infiltration in tumors. However, when we measured the levels of CD4+ and CD8+ cells in the tumor, both decreased very rapidly after administering OXi4503 (Figure 3). This decrease was probably the result of a significant drug-induced reduction in tumor blood flow as has been recorded with this drug [42]. The levels actually started to recover around 3–4 days after treatment, a time when the blood flow reductions are beginning to return to normal. Surprisingly, at later time intervals, the levels of CD4+ and CD8+ again declined, but we can offer no explanation for this effect.

Regardless of the mechanism, our results show the potential of VDAs to convert “immunological cold” tumors into ones that were “immunologically hot” and this may have clinical implications. There are a number of VDAs in clinical evaluation [32], but whether the effects we see with these combretastatin drugs is a general trend for all VDAs is not known and certainly worthy of further investigation. There are no examples in the literature showing VDAs to be capable of inducing tumor control, so the clinical potential of VDAs will be when used in combination with other more conventional therapies, especially radiotherapy. Indeed, the combination of VDAs and radiation is one of the most extensively investigated in pre-clinical studies [25,30]. There are also a number of pre-clinical studies showing that radiation and checkpoint inhibitors can be an effective combination in tumors [16,17,18]. Thus, combining VDAs with radiation and checkpoint inhibitors could be an effective therapy approach. As such, VDAs could be an excellent addition to some of the almost 200 clinical trials already ongoing with radiation and checkpoint inhibitors [31]. Of course, any future pre-clinical studies with VDAs, radiation, and checkpoint inhibitors would require assessment of potential radiation-induced normal tissue damage to compare with the tumor response to demonstrate a therapeutic benefit, similar to what we have done previously for VDAs and radiation [39,41,60,61]. There is also the issue of possible systemic toxicity resulting from treatment with the checkpoint inhibitors alone or when combined with the VDAs. This is especially true for anti-CTLA-4 in which any increase in T-cells could lead to attacks on normal tissues; anti-PD-1/PD-L1 are basically tumor specific in their action [5]. However, we found no evidence of any additional body weight loss when comparing the checkpoint inhibitor treated groups with the relevant non-checkpoint inhibitor group (Figure 4).

One of the most surprising findings from our study concerns the number of animals that actually showed an increased response to OXi4503 and anti-CTLA-4 (Figure 2). At the 25 and 50 mg/kg doses of OXi4503, some 40–50% of mice also receiving anti-CTLA-4 showed a TGT5 value outside the range seen with either OXi4503 dose alone. However, the remaining mice receiving OXi4503 and anti-CTLA-4 had a TGT5 value that was identical to that seen with OXi4503 alone. Why only some tumors responded, but not all, is unclear, especially since the mice were genetically similar, from the same birth period, implanted with tumor material from the same source at the same time, and treated with the same stock solutions of drugs. The differences in response is clearly an intriguing observation that warrants further investigation and may help explain why some patients respond to checkpoint inhibitors, yet many do not.

## 4. Materials and Methods

### 4.1. Animal and Tumor Model

Experiments involved using 13–19 week-old male and female CDF1 mice (Janvier Labs, France) with a C3H mammary carcinoma implanted in the right rear foot. Mice, kept four to a cage, had food and water ad libitum and a stable circadian rhythm secured with a light/dark interval of 12/12 h. The C3H mammary carcinoma model is an anaplastic adenocarcinoma that arose spontaneously in a C3H mouse at our institute and originally designated as HB [33]. The name was changed to the C3H mammary carcinoma when grown in the more stable CDF1 mouse variant [34]. Experimental tumors were produced following sterile dissection of large flank tumors as previously described [34]. Basically, every three months tumor material stored in liquid nitrogen was thawed and implanted on the flanks of mice. When large, these tumors were excised, then minced with a pair of scissors under sterile conditions. Tumor material could then be implanted either on the flanks of additional mice for continued passage or 5–10 μL of this material injected into the right rear foot of mice used for experimental studies. Experiments were initiated when tumors had reached approximately 200 mm^3^ in size (day 0). This volume was achieved two-three weeks after inoculation and was calculated from the formula D1 × D2 × D3 × π/6, where the D values represent the three orthogonal diameters measured with calipers. Attempts were made to randomize the tumor bearing mice into the different treatment groups. However, since the tumors grew at different rates they did not achieve the 200 mm^3^ starting volume at the same time. Consequently, some selection was necessary to ensure that tumors starting treatment on the same day were distributed among the different treatment groups. All animal studies were conducted according to the animal welfare policy of Aarhus University (http://dyrefaciliteter.au.dk), and with the Danish Animal Experiments Inspectorate’s approval. A schematic illustration of the experimental design is shown in Figure 5.

### 4.2. Drug Preparation and Treatment

The VDAs were Combretastatin A-4 phosphate (CA4P) and its A-1 analogue OXi4503; both kindly supplied by Mateon Pharmaceuticals (South San Francisco, CA, USA). The checkpoint inhibitors were anti-CTLA-4 antibody (invivoMab anti-mouse CTLA-4[CD152], clone 9H10), anti-PD-1 antibody (invivoMab anti-mouse PD-1[CD279], clone RMP1-14), and anti- PD-L1 antibody (invivoMab anti-mouse PD-L1[B7-H1], clone 10F.9G2), and all three were purchased from NordicBioSite/BioXCells. All drugs were freshly prepared before each experiment by dissolving in sterile saline (0.9% NaCl) and intraperitoneally (i.p.) injected into mice at a constant injection volume of 0.02 ml/g mouse body weight. Drug treatments were started when tumors reached 200 mm^3^ (day 0) and involved injecting CA4P (100 mg/kg) or OXi4503 (5–50 mg/kg) on days 0, 3, 7, and 10, with the latter shown to be the optimal application [42]. For the checkpoint inhibitors, a standard dose of 10 mg/kg was administered on days 1, 4, 8, and 11 regardless of the antibody. This schedule was designed to mimic experimental settings used by others [16,62].

### 4.3. Tumor Response Assessment

Tumors were measured daily, and when the tumor reached a size of 1200 mm^3^ the mouse was terminated by cervical dislocation. The response endpoint was tumor growth time (TGT5; time for tumors to reach 5 times the starting treatment volume). However, in some of our experiments, the tumor completely regressed, and in that situation, we continuously examined the mouse twice a week, and after approximately 3 months (90 days) the experiment was stopped, and the mouse sacrificed by cervical dislocation. We recorded mouse body weight to assess toxicity and, due to ethical issues, mice were terminated if they did not thrive or lost more than 20% in body weight.

### 4.4. CD4 and CD8 Immunohistochemistry

Mice treated with OXi4503 (50 mg/kg) were terminated by cervical dislocation between 1 to 10 days after VDA treatment, and their tumors excised, formalin-fixed, and paraffin embedded. Two serial 3 μm sections were cut, mounted on Superfrost Plus slides (Thermo Fisher Scientific, USA), and dried (60 °C, 1 h). They were stained firstly for CD4 and secondly for CD8 by Discovery Ultra (Ventana Medical Systems, Tucson, USA). Standard settings were used for deparaffinization and rehydration. The antigen retrieval was performed in CC1 at 100 °C for 32 min followed by endogenous peroxidase blocking by Chromomap inhibitor for 12 min. The first sections were incubated with rabbit monoclonal CD4 (1:100; EP19514, Abcam) and incubated for 32 min at room temperature. The secondary antibody, Omnimap anti-rabbit, was incubated for 16 min at room temperature. The second section was incubated with rabbit monoclonal CD8 (1:2000; EPR20305, Abcam) and incubated for 32 min at room temperature. The secondary antibodies, Discovery anti-Rabbit HQ and Discovery anti-HQ hrp, were both incubated for 16 min at room temperature. The stains were visualized with Discovery Chromomap DAB kit, and counterstained with hematoxylin and bluing reagent. Visiopharm Integrator System 2019.02.1.6005 (Visiopharm A/S, Hørsholm, Denmark) was utilized for automated quantification of whole slide images captured at 20× magnification by Nanozoomer 2.0HT (Hamamatsu Phototonics K.K., Hamamatsu City, Japan). Regions of interest (the entire lesion and tumor areas) were manually outlined, with necrotic tissue excluded. Automated quantifications within regions of interest were based on thresholding of the brown staining color (DAB), which was highlighted by a color deconvolution step. Features of the red, green, and blue color levels enhanced the remainder tissue. Different post-processing algorithms (primarily morphological operations and changes by area or surrounding) subsequently improved thresholding results. All labels of image analysis were manually reviewed, and any considerable errors were omitted from the analysis (e.g., tissue folds or unspecific dab staining). CD4+ and CD8+ percentage levels were calculated by dividing their area of cytoplasmatic staining with the total tissue area of the region of interest.

### 4.5. Statistical Analysis

Results are presented as the mean (± 1 standard error of the mean) when normally distributed or as individual values with medians if not. A Student’s *t*-test or a Wilcoxon–Mann–Whitney test used for statistical analyses, with a *p*-value < 0.05 considered statistically significant for both tests.

## 5. Conclusions

The recent success of immunotherapy in improving outcome in cancer patients has resulted in this treatment option being considered the fourth modality in cancer treatment after surgery, radiotherapy, and chemotherapy. However, despite this success, many patients are unresponsive to immunotherapy. Our current pre-clinical study found that clinically relevant VDAs could be an effective treatment option for combining with immunotherapy to convert immune unresponsive tumors in to immune responders. Translating this approach in to clinical evaluation, could result in more patients benefitting from immunotherapy treatment.

## Figures and Tables

**Figure 1 ijms-21-04778-f001:**
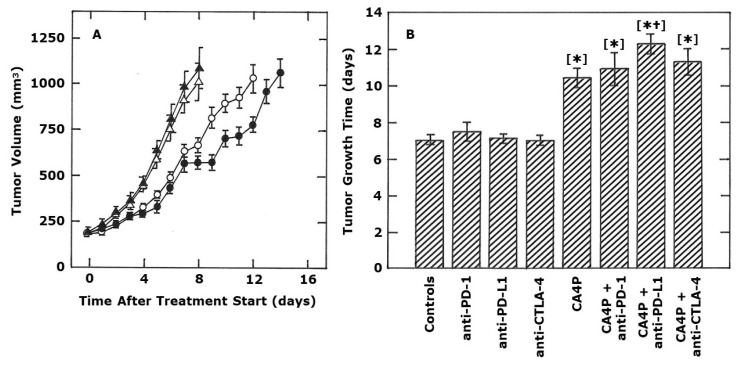
Effect of CA4P and monoclonal antibody inhibitors of PD-1, PD-L1, and CTLA-4 on the growth of a C3H mammary carcinoma implanted in CDF1 mice. Representative growth curves are shown in (**A**) for control mice (open triangles), or mice intraperitoneal (i.p.) injected with 10 mg/kg anti-PD-L1 antibody (closed triangles), 100 mg/kg CA4P (open circles), or CA4P and anti-PD-L1 combined (closed circles). (**B**) shows the tumor growth time (time for tumors to reach 5 times the starting treatment volume) for control mice, or mice i.p. injected with anti-PD-1, anti-PD-L1, or anti-CTLA-4 (all at 10 mg/kg), and/or CA4P (100 mg/kg). For both figures, treatments were started when tumors had reached a volume of 200 mm^3^ (day 0). The actual treatment days were 0, 3, 7 and 10 (CA4P) or 1, 4, 8 and 11 (checkpoint inhibitors). All results show means (± SEM) from at least 7 mice/group. Statistical comparisons of the data in (**B**) were made using a Student’s *t*-test and show those groups that were significantly different (*p* < 0.05) from controls [*] or CA4P alone [†].

**Figure 2 ijms-21-04778-f002:**
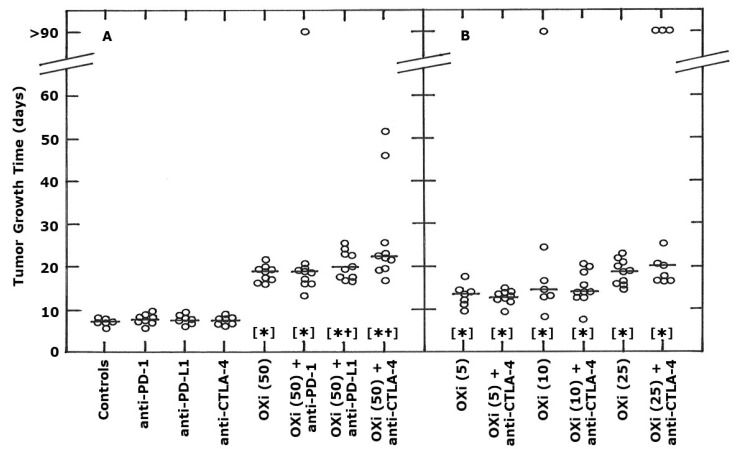
Effect of OXi4503 (OXi) and monoclonal antibody inhibitors (10 mg/kg) of PD-1, PD-L1, and CTLA-4 on the growth of a C3H mammary carcinoma implanted in CDF1 mice. These mice were i.p. injected with different drugs, with the treatments starting when tumors had reached a volume of 200 mm^3^ (day 0). The actual treatment days were 0, 3, 7 and 10 (OXi) or 1, 4, 8 and 11 (checkpoint inhibitors). Results are individual values with the line indicating the median for each group and show the tumor growth time (time for tumors to reach 5 times the starting treatment volume). Values at >90 days indicate those tumors that were controlled so no actual tumor growth time value was possible. For (**A**), the results are for control animals, mice treated with each checkpoint inhibitor (10 mg/kg) alone, a high OXi dose (50 mg/kg) alone, or OXi and each inhibitor combined. (**B**) shows results using lower OXi doses (5, 10, or 25 mg/kg) with/without anti-CTLA-4. The different OXi doses are shown in the parentheses on the x-axis. Statistical comparisons of the data in both figures were made using a Wilcoxon-Mann-Whitney test and show those groups that were significantly different (*p* < 0.05) from controls [*] or each respective OXi dose alone [†].

**Figure 3 ijms-21-04778-f003:**
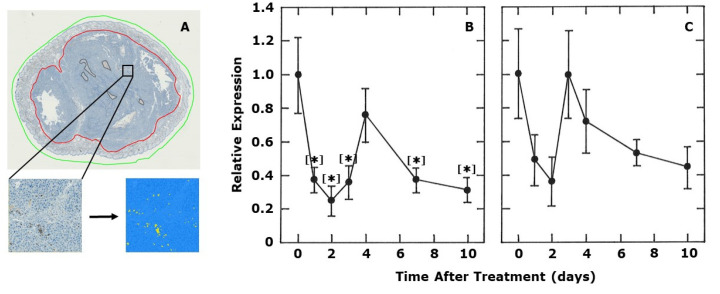
The effect of OXi4503 on tumor infiltrating lymphocytes. A representative tumor section to illustrate how the immune cells were quantified is shown in (**A**). The green line marks the entire tumor section, while the red line indicates the gross tumor area. Areas defined by the brown lines are necrotic regions, which are subtracted from the total tumor area in order to define the net tumor area. The brown dab-stained cells on the tumor section or in the enlarged subunit are the immune cells (CD4+ cells in this example). This image is automatically converted to a binary image, with the immune cells appearing yellow among the blue colored HE-stained cells as shown. Quantitative estimates of the relative levels of CD4+ (**B**) or CD8+ (**C**) determined from immunohistochemistry are shown as a function of time after treating 200 mm^3^ C3H mammary carcinoma bearing CDF1 mice with a single dose of OXi4503 (50 mg/kg). Results show means (± SEM) for at least 6 mice/group. Statistical comparisons were made using a Student’s *t*-test and show those groups that were significantly different (*p* < 0.05) from controls [*].

**Figure 4 ijms-21-04778-f004:**
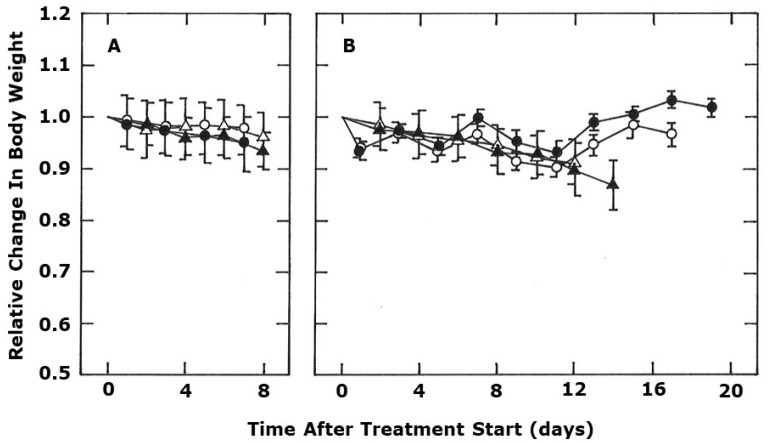
The effect of different treatments on the body weight of C3H mammary carcinoma bearing CDF1 mice. Measurements were started when tumors had reached 200 mm^3^ (day 0) and show body weight changes relative to the starting value. In (**A**), the symbols are for control mice (open triangles), or mice treated with either anti-PD-L1 (closed triangles), anti-PD-1 (open circles), or anti-CTLA-4 (closed circles). (**B**) shows the effect of treating mice with either CA4P (open triangles), CA4P + anti-PD-L1 (closed triangles), OXi4503 (open circles), or OXi4503 + anti-CTLA-4 (closed circles). The actual treatments were CA4P (100 mg/kg) or OXi4503 (50 mg/kg) i.p. injected on days 0, 3, 7, and 10, or the checkpoint inhibitors (10 mg/kg) i.p. injected on days 1, 4, 8, and 11. All results show means (+ 1 SEM) from at least 7 mice/group. A Student’s *t*-test was used to compare each checkpoint inhibitor with controls, or each VDA with the corresponding VDA + checkpoint inhibitor, and no significant (*p* < 0.05) differences were found.

**Figure 5 ijms-21-04778-f005:**
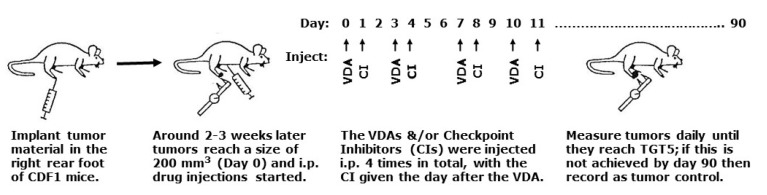
Schematic illustration of the overall experimental design.

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
