# Peer review of "Tumors Resistant to Checkpoint Inhibitors Can Become Sensitive after Treatment with Vascular Disrupting Agents"

_ijms, 2020, doi:10.3390/ijms21134778_

Round 1
Reviewer 1 Report
The manuscript by Horsman et al. states that using vascular disrupting agents (VDAs) along with checkpoint inhibitors could effectively restrict C3H mammary carcinoma tumor growth. They further suggest that combining VDAs with radiation and checkpoint inhibitors could be an effective therapeutic approach for various tumors. Though these approaches could be beneficial for restricting tumors, there are several concerns regarding this manuscript:
- What was the age group of the mice used for their experimental study, as that could be a contributing factor in the immune response?
- Why a Student’s T-test or a Wilcoxon-Mann-Whitney test was used while comparing multiple parameters instead of ANOVA?
- The figures do not show any statistics though the authors mention in the materials and methods section. Also, in figure 2 the number of n is different for the different treatment groups.
- The authors report “CA4P (100 mg/kg), administered twice weekly for two weeks, significantly increased the tumor growth time (TGT5; time for tumors to reach 5 times the starting treatment volume) and this was further significantly increased when CA4P was combined with anti-PD-L1, but not when the VDA was combined with the other two antibodies”. This is not reflected in the figure.
- Figure 2 and its description is very confusing and difficult to follow. Is there any dose-dependent effect of Oxi4503? Why the same inhibitors were not used at different doses of Oxi4503?
- The authors have over speculated their data, which is also reflected in their discussion. They talk about the hyperthermia effect, hypoxia effect, and radiation therapy effect and thereby try to correlate the VDAs therapeutic potential in treating the tumors. The authors do not have enough data to support these facts. This makes the discussion misleading and diverting from the authors results and data shown. Therefore, I would rather recommend the authors to base their hypothesis on the current finding and be precise in their discussion. Also, the authors should try to avoid any assumptions for which strong evidence is not shown in the results section. For example, how this approach could be used for all forms of tumors?
- There are some grammatical errors. For example, Line 300-301, etc. Therefore, I would recommend the authors to have a person proofread the manuscript whose native language is English.
Author Response
- We agree with the reviewer that age could be a contributing factor in influencing immune response. However, in our study the mice were all of a similar age at the time of treatment, specifically 13 to 19 weeks old, so age is unlikely to influence the differences in immune response we observed. In retrospect, we should have actually stated the mice age in the original submission, and now we clearly state the mice age in the materials and methods section (please see line 327).
- We did not actually compare multiple parameters as such. Instead, it was a simple comparison between two independent parameters each time (i.e., Controls vs. CA4P alone or OXi4503 alone vs. OXi4503 + anti-CTLA-4) and as such we felt that a Student’s T-test (where the data was normally distributed) or a Wilcoxon-Mann-Whitney test (where the data was not normally distributed) was more appropriate than ANOVA.
- We have now added statistics to all figures and the figure legends to clarify this issue. It is true that the number of “n” is different for the different treatment groups in figure 2. We performed each set of experiments twice, but each time we tried to include all treatment groups in one experiment. Since we had a limit as to the number of tumor bearing mice we could set-up each time, we had to be slightly selective in the number of animals in each group. So, those groups where we had data showing no effect (i.e., controls, or antibodies alone; we had data for these from figure 1 and figure 2) we decided to have less mice per group and instead include more animals in those groups where we expected changes.
- The data of figure 1 does suggest that the effect of CA4P is enhanced with all three antibodies. However, only for the combination of CA4P and anti-PD-L1 is there a statistically significant difference to the TGT5 value obtained with CA4P alone; no statistical significant increase above that seen with CA4P alone was found when we treated with CA4P + anti-PD-1 or anti-CTLA-4. This was described in the text, but in line with the reviewer’s comment we have modified the text (please see lines 102-105), and to further clarify the situation on the figure we have now added symbols to show which changes are significant or non-significant.
- We can understand the reviewer’s difficulty with understanding figure 2 and its description. We have thus modified our comments (please see lines 118-138) to try and improve the understanding, especially concerning the dose dependent effect of OXi4503. As to the comment as to why we did not combine all the different OXi4503 doses with all three checkpoint inhibitors, rather than just anti-CTLA-4, then we agree that for completeness perhaps this should have been done. However, there was little indication from our results with the highest OXi4503 (50 mg/kg) dose and anti-PD-1 or anti-PD-L1 to suggest any effect would be seen using lower OXi4503 doses. Since the antibodies are very expensive and in trying to keep to the 3R’s for animal welfare we felt that doing these additional studies would be unnecessary. Some comments to this are now added to the text (please see lines 134-136).
- In response to the reviewer’s comments, then we have significantly changed the discussion and focused more on our findings and less on other speculative issues. In that context, we have removed the discussion of the hyperthermia effect, significantly reduced our discussion of the hypoxia issue (we feel this is still as important issue based on our knowledge about the mechanism of VDAs), and removed comments about other tumor models. As to radiation therapy, then this was specifically included at the request of the Editor of this special issue (Prof. Gaipl). The focus of that special issue is radiation therapy and when we were asked to submit a manuscript, I told him that we did not have sufficient radiation data for a full manuscript, but had data on VDAs with immune therapy. He decided he wanted us to submit that study, but try to include as much reference to radiation as possible, especially since the clinical application of VDAs is most likely in combination with radiation. As such, we believe that the discussion of possible radiation aspects should remain.
- The reviewer is correct that there are some grammatical errors. To be honest this was an oversight on my side. As first author, I was responsible for checking the language, and although I may have a Scandinavian sounding last name and live in Denmark, I am actually English, being born and raised in London, and should have been more careful checking the sections written by my Scandinavian co-authors! I have now carefully checked the English language.
Reviewer 2 Report
The study entitled “Tumors resistant to checkpoint inhibitors can become sensitive after treatment with vascular disrupting” by Horsman et al. investigated the combinatorial effect of vascular disrupting agents such as CA4P and its analogue OXi4503 with checkpoint inhibitors antibodies anti-PD-1, PD-L1, or CTLA-4 antibodies to convert an unresponsive immune cold-tumor into an immune hot-spot responder.
The authors show that although the checkpoint inhibitor antibodies were not effective alone some of these had greater effect when combined with both C4AP and OXi4503 suggesting their application in a pre-clinical and clinical setting. There are other studies that have shown that CA4P may enhance the activity of anti-CTLA4, anti-PD1, and anti-PD-L1 antibodies. However, these studies not mentioned. The study is well conducted and presented.
In Fig 1. Combinations with anti-CTL4A and anti-PD-1 are slightly increased when compared to C4AP alone. The author should correct their statement on line 102-103. Schematics showing the experimental design and dose details should be included.
The statistical analysis should be shown in the figure and the legend should mention what test was used to calculate the significance.
Baseline tumor value should be included and also figure showing the change in tumor volume over days and different treatment will be more informative and should be included.
The authors measured the mouse weight was recorded but not mentioned anywhere in the result. This is important to understand the toxicity of the different substances that were evaluated in this study.
Fig 3 would benefit including the immunohistochemistry images to present the differences in the number of CD4+ and CD8+ cells. What is the reason of the decrease in the number of CD4 and CD8 T cells at 7-10.
The discussion is lengthy and can be shortened. The mechanisms of action of each agent used is different thus the authors should discuss their results in the context of these agents and their probable mechanism in combinations.
Author Response
The reviewer mentions that there are other studies that have shown that CA4P may enhance the activity of anti-CTLA-4, anti-PD-1, and anti-PD-L1, but these studies are not mentioned by us. It was never our intention to ignore the work of others, especially studies focusing on the same issues as we have, and we would gladly include those other references. Unfortunately, we have absolutely no idea as to what these other studies are. Previously, Mateon pharmaceuticals, the company that has CA4P, had approached us and another research group to do the initial testing of CA4P with checkpoint inhibitors; it was based on some preliminary unpublished data they had obtained elsewhere. I have since contacted that other research group and they inform me that they have not published their findings. We also checked on PubMed using identifiers like “combretastatin/CA4P” and “immune therapy/checkpoint inhibitors/anti-CTLA-4/anti-PD-1/anti-PD-L1”, and for all the references that showed up none actually involved the combination referred to by the reviewer. Of course, the data in question might be “hidden” in other publications or published in Journals not readily available and thus missed. If the reviewer would be kind enough to give the relevant references then we would of course include them.
The data of figure 1 does suggest that the effect of CA4P is enhanced with all three antibodies. However, only for the combination of CA4P and anti-PD-L1 is there a statistically significant difference to the TGT5 value obtained with CA4P alone; no statistical significant increase above that seen with CA4P alone was found when we treated with CA4P + anti-PD-1 or anti-CTLA-4. This was described in the text, but to further clarify the situation we have modified the text (please see lines 102-105), and added symbols to the figure to show which changes are significant or non-significant. The doses used in all experiments are actually listed in the legend of each figure. As to a schematic showing the treatment schedules then a new figure has been made and included (Figure 5).
The statistical analysis is now added to all figures and the relevant statistical test used for the data shown in each figure is now included in the legend of each figure.
Baseline tumor volumes were the same throughout (200 mm3); this is stated in the Material and Methods section (line 338) and in each figure legend, so showing these values in each figure would we believe make each figure very complicated and confusing. However, a figure showing the change in tumor volume over days for the most significant treatments in figure 1 has now been added to this figure (Figure 1A) as requested.
We agree with the reviewer that it is important to understand the toxicity of the different treatments. Thus, to illustrate this we have now included an extra figure showing the relative change in mouse body weight as a function of time (Figure 4) as requested, with comments to this figure in the results (lines 185-194) and discussion (lines 309-314) sections.
As requested, we have also included an example of the immunohistochemistry images and how they were used to calculate CD4+/CD8+ cell infiltration; this has been added to figure 3 (please see figure 3A). We have absolutely no explanation as to why there is a second drop in the number of CD4 and CD8 cells at 7 and 10 days. However, we do add some comments to this issue on lines 292-293.
We have considerably changed the discussion. This includes deleting the discussion of the hyperthermia effect, significantly reducing our discussion of the hypoxia issue (we feel this is still as important issue based on our knowledge about the mechanism of VDAs), removing speculative comments about other tumor models, and focusing on possible mechanisms as requested (please see lines 226-293).
Round 2
Reviewer 1 Report
I thank the authors for satisfactorily addressing all the concerns.
Reviewer 2 Report
The is the second submission for this manuscript and the authors have responded to the concerns raised by the previous review panel.